# Impact of a Carbohydrate Mouth Rinse on Corticomotor Excitability after Mental Fatigue in Healthy College-Aged Subjects

**DOI:** 10.3390/brainsci11080972

**Published:** 2021-07-23

**Authors:** Stephen P. Bailey, G. Keith Harris, Kaitlin Lewis, Tracy A. Llewellyn, Ruth Watkins, Mark A. Weaver, Bart Roelands, Jeroen Van Cutsem, Stephen F. Folger

**Affiliations:** 1Department of Physical Therapy Education, Elon University, Elon, NC 27244, USA; klewis25@elon.edu (K.L.); tllewellyn@elon.edu (T.A.L.); folgers@elon.edu (S.F.F.); 2Department of Food, Bioprocessing, and Nutrition Sciences, NC State University, Raleigh, NC 27695, USA; gkharris@ncsu.edu (G.K.H.); rhwatkin@ncsu.edu (R.W.); 3Department of Mathematics & Statistics, Elon University, Elon, NC 27244, USA; mweaver11@elon.edu; 4Human Physiology and Sports Physiotherapy Research Group, Vrije Universiteit Brussel, 1050 Brussels, Belgium; bart.roelands@vub.be (B.R.); Jeroen.van.cutsem@vub.be (J.V.C.); 5Sport and Exercise Science, School of Psychology and Life Sciences, Canterbury Christ Church University, Canterbury Campus, Canterbury, Kent CT1 1QU, UK; 6VIPER Research Unit, Royal Military Academy, 1000 Brussels, Belgium

**Keywords:** TMS, mental fatigue, carbohydrate

## Abstract

Mental Fatigue (MF) has been associated with reduced physical performance but the mechanisms underlying this result are unclear. A reduction in excitability of the corticomotor system is a way mental fatigue could negatively impact physical performance. Carbohydrate (CHO) mouth rinse (MR) has been shown to increase corticomotor excitability. PURPOSE: The purpose of this study was to determine if CHO MR impacts corticomotor excitability after MF. METHODS: Fifteen subjects (nine females, six males; age = 23 ± 1 years; height = 171 ± 2 cm; body mass = 69 ± 3 kg; BMI = 23.8 ± 0.7) completed two sessions under different MR conditions (Placebo (PLAC), 6.4% glucose (CHO)) separated by at least 48 h and applied in a double-blinded randomized fashion. Motor-evoked potential (MEP) of the left first dorsal interosseous (FDI) was determined by transcranial magnetic stimulation (TMS) before and after MF. Perceived MF was recorded before and after the MF task using a 100 mm visual analog scale (VAS). RESULTS: MF was greater following PLAC (+30.4 ± 4.0 mm) than CHO (+19.4 ± 3.9 mm) (*p* = 0.005). MEP was reduced more following PLAC (−16.6 ± 4.4%) than CHO (−3.7 ± 4.7%) (*p* < 0.001). CONCLUSIONS: CHO MR was successful at attenuating the reduction in corticomotor excitability after MF. Carbohydrate mouth rinse may be a valuable tool at combating the negative consequences of mental fatigue.

## 1. Introduction

Mental fatigue is a psychobiological state caused by prolonged periods of demanding cognitive activity [1], which can be observed using subjective, behavioral, and physiologic measures. Mental fatigue has been shown to precipitate detrimental effects on endurance, motor skill, decision-making performance, and perception of effort, but is not associated with changes in maximal force production. For example, Filipas and colleagues [2] found power output and pedal cadence to be reduced during a cycling time trial in sub-elite cyclists when they were mentally fatigued. In contrast, Rozand and associates [3] found no change in maximal voluntary contraction (MVC) or contraction force during a supramaximal electrical stimulation of the knee extensors when subjects were mentally fatigued, suggesting that mental fatigue does not influence the voluntary activation of muscle contraction.

While the specific mechanisms that precipitate a reduction on physical performance following mental fatigue remain unclear, several authors have associated this phenomenon with changes in cortical activity [4,5,6]. For example, Piers and colleagues observed a decrease in cycling performance and an increase in theta band activity in the prefrontal cortex following a mentally fatiguing task in recreational cyclists [6]. Van Cutsem and colleagues have suggested that alterations in motor control may force mentally fatigued subjects to increase their central motor command and muscle recruitment to maintain physical performance. Increasing the intensity of central motor command needed to facilitate movement could subsequently trigger an increase in perception of effort [1]. Perceived exertion during exercise has frequently been found to be elevated following mental fatigue [7,8], and the positive relationship between perceived effort and central motor command has been well established [9,10]. The need for an increased central motor command to overcome a mentally fatigued state is supported by the work of Pageaux and colleagues who found mental fatigue to be associated with an increased EMG activity in the vastus lateralis during cycling [11]. These findings suggest that central motor command and muscle recruitment may have to be increased, in order to maintain power output in a mentally fatigued state.

Corticomotor excitability is often used to evaluate the impact of central motor command on physical function or performance. Changes in electrical activity of the motor cortex have been associated with changes in corticomotor excitability as measured using transcranial magnetic stimulation (TMS) [12,13]. Using TMS several authors have documented the relationship between corticomotor excitability and motor performance and function in healthy [14,15,16] and diseased [17,18] populations. Moscatelli and associates have recently described the value and limitations of using TMS to understand the role of motor cortex excitability in physical sport performance [19]. These authors contextualized how TMS can be used to understand function of the corticospinal tract in “top-down” paradigms. While mental fatigue has been associated with a reduction in physical performance, the impact of mental fatigue on corticomotor excitability remains unclear.

The Stroop Color Word Test (SCWT) is perhaps the most common tool used to induce mental fatigue in the laboratory. Investigators typically require subjects to participate in the SCWT for varying lengths of time (30 to 90 min) and document changes in perceived mental fatigue as a result. The SCWT challenges the anterior cingulate cortex (ACC) and its altered function has been associated with impairment of cognitive control, response preparation, and motor control subsequent to mental fatigue [20]. The ACC receives input from and sends information to the primary motor cortex, which gives rise to corticospinal projections that drive motor function [21]. Tanaka et al. [22] demonstrated that mental fatigue suppresses the activity of the right ACC altering motor function, primarily by increasing movement duration and slowing response time.

Previous investigations have demonstrated that corticomotor excitability can be enhanced following application of a carbohydrate (CHO) mouth rinse [23]. The mechanism underlying this improvement in performance is unclear. However, it is suggested that the CHO mouth rinse enhances the excitability of the motor cortex via stimulation of oral carbohydrate receptors. Gant and colleagues provided supportive evidence for this premise when they found the motor evoked potential (MEP) response to transcranial magnetic stimulation (TMS) in the biceps muscles was increased by 30% after a CHO mouth rinse [23]. Mental fatigue can occur in various sport and industrial settings that also require physical performance. Non-pharmacological and accessible strategies, like CHO mouth rinse, could be useful in maintaining physical performance when someone is at risk for mental fatigue.

The purposes of this investigation are to determine; (1) if mental fatigue precipitates a change in corticomotor excitability; and (2) if the effects of mental fatigue can be counteracted with the use of a CHO mouth rinse. It is hypothesized that the SCWT will induce mental fatigue by decreasing corticomotor excitability, but a CHO mouth rinse will counteract this effect.

## 2. Materials and Methods

### 2.1. Participants

Fifteen College-aged (9 female, 6 male; age = 23 ± 1 years; height = 171 ± 2 cm; body mass = 69 ± 3 kg; BMI = 23.8 ± 0.7) physically active (3 or more exercise sessions per week lasting at least 30 min) individuals in good health participated in this investigation. All subjects participating in this investigation were right hand dominant. Individuals were screened for appropriateness of participation in this investigation using the Transcranial Magnetic Stimulation Adult Safety Screen (TASS) created by Keen and associates [24]. Subjects were excluded from participation if they were unable to provide informed consent; treated clinically with non-invasive brain stimulation for a brain or psychological disorder within the past year; suffered a hemorrhagic stroke or had a history of cerebral hemorrhage or cerebral edema associated with the ischemic infarct requiring treatment; displayed symptoms of altered upper extremity function or sensation; had any implanted metallic medical devices; pregnant or trying to become pregnant; had a history of significant mental illness, a diagnosis of/or family history of epilepsy; suffered from uncontrolled hypertensive or hypotensive, or had any scalp wounds or infections. The sample size for this investigation was estimated from published data from our laboratory describing the expected change in corticomotor excitability following application of a maltodextrin carbohydrate mouth rinse (18%) [25], and the within session variability in corticomotor excitability (429 μV) we observed during pilot data collection for the specific muscle group examined in this investigation. Using this data, a sample size of 15 subjects was estimated to reach an appropriate statistical power (>0.80) [26]. Prior to participation, subjects read and signed a consent form approved by the University’s Institutional Review Board for Protection of Human Subjects in Research.

### 2.2. Experimental Design

Subjects reported to the laboratory on 3 occasions with each session separated by at least 48 h. Subjects were asked to not eat or drink anything for the 4 h prior to each session and to refrain from heavy exercise during the 24 h preceding each session. The initial session lasted approximately 60 min, during which time subjects completed the transcranial magnetic stimulation (TMS) adult safety screen (TASS) for contraindications to TMS [27] and were familiarized with all experimental procedures including TMS stimulation to produce the targeted muscle activity.

Following the familiarization session, two experimental sessions occurred at approximately the same time of day for each subject. During these sessions, corticomotor excitability was assessed by measuring the motor-evoked potential (MEP) of the left first dorsal interosseous (FDI). MEP of the FDI was evoked at rest by TMS, before and immediately after completion of a task designed to cause mental fatigue. During one session subjects used a carbohydrate (CHO) mouth rinse and during the other they used a placebo (PLAC) mouth rinse. Mouth rinse treatments were applied in a randomized, counter-balanced order. The subjects and the individuals collecting data were blind to the nature of the mouth rinse applied during each session. The experimental protocol for each session is displayed in Figure 1. During all testing subjects were seated comfortably in a Boyd Industries (Clearwater, FL, USA) E3010LC exam and treatment chair. While seated in the chair a footstool was used to maintain 90° flexion at the hips, knees, and ankles. The head was supported by a headrest and the left arm was placed comfortably (neutral shoulder flexion, slight shoulder abduction, and 90° elbow flexion) on a Hill-Rom Art of Care 365 adjustable table.

### 2.3. Mouth Rinse

The two mouth rinse conditions were (1) a placebo (PLAC), and; (2) a 6.4% maltodextrin carbohydrate (CHO) solution. The mouth rinses were created to have the same sweetness based on dextrose equivalents by adding artificial sweetener (1.6% Stevia-based). Coloring was also added to each mouth rinse so that they were indistinguishable in color. Upon formulation, the taste similarity of the two mouth rinses was confirmed in a novel group of subjects (n = 8) who did not participate in this investigation. Each MR was 25 mL in volume and delivered to the subject in a semi-opaque polypropylene tube. When rinsing, subjects were instructed to place the fluid in their mouth and hold it for 20 s and then return the rinse back to the delivery tube. A mouth rinse was applied to subjects between each bout of the SCWT that occurred during the trial (Figure 1). As a result, subjects completed 5 “fresh” mouth rinses of the same treatment (PLAC or CHO) during each session. The length of time between each bout of SCWT was approximately 30 s.

### 2.4. Corticomotor Excitability

Corticomotor excitability was evaluated using a Magstim 200 transcranial magnetic stimulator (TMS) (Magstim, West Wales, UK) equipped with a D-70 Remote Flat figure-8 cone shaped coil. During all trials, electromyography (EMG) was constantly measured. EMG electrodes (Ag-AgCl, 10-mm diameter) (Thought Technology Ltd., Montreal West, QC, Canada) were placed on the radial side of the base of the second proximal phalanx (index finger), belly of the 1st dorsal interosseous (FDI) muscle, and the ulnar styloid process after the area was cleaned with alcohol. EMG was sampled at 2048 Hz using band (1–1000 Hz) and notch (60 Hz) filters.

#### 2.4.1. Determination of “Hot Spot”

Prior to each trial, the location, or “hot spot”, where TMS created the maximal evoked potential (MEP) for the FDI was mapped and documented using the ANT Visor 2 Neuronavigation System (Hengelo, The Netherlands). This real-time position tracking system facilitates consistent stimulator positioning during each experimental session. To identify the “hot spot”, subjects relaxed their left hand while the researcher stimulated over the hand representation area of the contralateral motor cortex. The stimulus location that produced the highest FDI motor evoked potential (MEP) amplitude was identified as the “hot spot”. In order to identify the resting motor threshold (RMT), the intensity delivered by the TMS stimulation was then adjusted to identify the lowest magnitude that creates a clear visible (>50 μV) MEP above background EMG in 3 out of 5 stimuli.

#### 2.4.2. Determination of Corticomotor Excitability

When evaluating motor excitability, subjects again relaxed their left hand, and TMS stimulation (120% of RMT) was delivered to the “hot spot” for the FDI. The TMS stimulation intensity used was similar for the mouth rinse treatments (PLAC = 53 ± 2%; CHO = 52 ± 2%). This process was repeated 10 times over a 1-min time period with 5 s between each stimulation. Peak to peak MEP (mV) for each repetition was averaged to determine the MEP at that time point. Within trial variability of MEP was 0.108 ± 0.066 μV or 5.58 ± 1.68%. Stimulations (4.8%) that did not produce a clear MEP were discarded and not used for data analysis.

### 2.5. Mental Fatigue

Mental fatigue was facilitated in subjects using the Stroop Color Word Task (SCWT) performed on a PC platform and delivered via Inquisit 4 (Millisecond Software, LLC, Seattle, WA, USA). Subjects completed six blocks of 336 stimuli each over an approximate 60 min period. During the SCWT, four colored words (Red, Green, Blue, and Black) were presented one at time on a computer screen. Subjects were required to indicate the color of the word while ignoring the meaning of the word. Each word was presented on the screen in a 34-point font for 1000 ms, with a variable inter-stimulus interval. Subjects were also occasionally provided a control stimulus that consisted of a red, green, blue, or black block and subjects were required to identify the color of the block. SCWT performance was evaluated by determining reaction time and accuracy of response for congruent (40%), incongruent (40%), and control (20%) stimuli. Subjects provided their input to the computer using a Cedrus RB740 response pad via their right hand. Perceptions of mental fatigue were assessed after the mental fatigue task using a continuous (100 mm) visual analog scale with anchors of no fatigue (0 mm) and extreme fatigue (100 mm).

### 2.6. Statistical Analyses

For the repeated measures ANOVAs, the assumption of multivariate normality was assessed using the Henze-Zirkler test [28]. When the Henze-Zirkler test suggested that the normality assumption was questionable (*p* < 0.05) the data were transformed using natural logarithms prior to further analysis. Two-way (Treatment × Time) repeated measures ANOVAs evaluated the effect of the mouth rinse on absolute and relative (change from pre-mouth rinse measurement) measures of cognitive performance and corticospinal excitability. SCWT data were analyzed using a linear mixed effect regression model that allowed for missing response data. These models included fixed effects for time, treatment, and the treatment-by-time interaction, along with corresponding random coefficients for these same effects for each subject. The model assumptions were reviewed by the plots of the Studentized residuals and normal probability plots for the Students residuals and predicted random effects. Using these methods, we identified a need to log transform the incongruent reaction time responses. Following the least significant difference approach, contrast tests were performed to assess the mean change over time within each treatment group if a significant interaction (treatment-by-time) was observed during the analysis of main effects. Data is presented as mean ± standard error (M ± SE). A significance level of *p* < 0.05 was set *a priori* for all analyses. All statistical analyses were performed using SAS, version 9.4 (SAS Institute, Cary, NC, USA).

## 3. Results

### 3.1. SCWT Performance

SCWT evaluated the reaction time and accuracy for congruent, incongruent, and control stimuli. No significant differences in mean accuracy were observed across mouth rinse condition or time for any of the types of stimuli (Table 1). No significant differences in mean reaction time were observed across the type of mouth rinse for congruent (F = 0.85, *p* = 0.373), incongruent (F = 0.32, *p* = 0.582), or control (F = 0.66, *p* = 0.429) stimuli. There were significant differences in mean reaction time across time for congruent (F = 61.79, *p* < 0.001), incongruent (F = 47.39, *p* < 0.001), and control (F = 29.91, *p* < 0.001) stimuli. Mouth rinse × time interactions were not significant for congruent (F = 0.17, *p* = 0.688), incongruent (F = 0.38, *p* = 0.546), or control (F = 0.12, *p* = 0.738) stimuli.

### 3.2. Mental Fatigue

Perceived mental fatigue data did not clearly violate the assumption of normality (*p* = 0.130) and as a result the original raw perceived mental fatigue data were used for statistical analysis. Mean absolute perceived mental fatigue was not significantly different across mouth rinse conditions (F = 0.280, *p* = 0.605). However, it was significantly different across time (F = 48.079, *p* < 0.001) and a mouth rinse × time interaction (F = 11.313, *p* = 0.005) (Table 2) was present. Perceived mental fatigue was not different between PLAC and CHO at Pre (F = 1.61, *p* = 0.226) or Post (F = 2.79, *p* = 0.117). In comparison, the increase in perceived mental fatigue was greater (F = 3.364, *p* = 0.005) following PLAC (30.4 ± 4.0 mm) than CHO (19.4 ± 3.9 mm) (Figure 2).

### 3.3. Corticomotor Excitability

Original MEP data did not meet the assumption of multivariate normality (*p* < 0.001); however, after log transformation MEP data did meet this assumption (*p* = 0.147). As a result, log transformed MEP data were used for statistical analysis. Mean MEP was significantly different across mouth rinse treatments (F = 5.93, *p* = 0.029) and time (F = 7.10, *p* = 0.019). The mouth rinse × time interaction was also significant (F = 21.98, *p* < 0.001) for MEP. There was no difference between groups in MEP prior to mental fatigue (F = 3.48, *p* = 0.083), but there was a difference observed between PLAC and CHO after mental fatigue (F = 6.54, *p* = 0.023). There was no evidence of change in average MEP from Pre (2.43 ± 0.41 mV) to Post (2.25 ± 0.43 mV) mental fatigue in CHO (F = 1.28, *p* = 0.278). In comparison, average MEP decreased significantly from Pre (1.92 ± 0.25 mV) to Post (1.57 ± 0.22 mV) in PLAC (F = 14.55, *p* = 0.002) (Figure 3). The relative reduction in MEP was also greater following PLAC (−16.6 ± 4.4%) than CHO (−3.7 ± 4.7%) (*p* < 0.001) (Figure 4).

## 4. Discussion

The first goal of this investigation was to determine whether mental fatigue had a negative impact on corticomotor excitability. In order to effectively assess this question, a state of “mental fatigue” must be simulated. The SCWT was used in this investigation to induce mental fatigue. This strategy has been used by many investigators who have asked participants to complete this test over periods of time ranging from 10 to 90 min [1,29,30]. Not all of these investigations reported changes in performance during the SCWT, but relatively consistent results have been seen in those investigations that did.

For investigations where the SCWT was applied over a 30–45 min period, performance of the SWCT typically remains unchanged [2,31]. In contrast, Van Cutsem and colleagues [31] required subjects to complete 8 blocks of the SCWT over a 90 min period and observed changes in SCWT performance. These authors observed an overall reduction in reaction time and a reduction in accuracy during incongruent stimuli when a placebo mouth rinse was used [31]. We observed a similar reduction in reaction time, but did not observe a reduction in accuracy. A potential explanation for these variant findings is the difference in the length of time the SCWT test that was applied, with Van Cutsem and colleagues [31] applied the test over 90 min as compared to the 60 min used in this investigation. Interestingly, while there was a 30-min difference in the length of time that the SCWT was applied, both investigations required subjects to respond to the same number of stimuli (2016). Besides the length of the test, there were some other small differences in the application of the protocol designed to induce mental fatigue. Van Cutsem and colleagues [31] required subjects to continue the SCWT during the mouth rinse, while subject in this investigation stopped the SCWT while they were completing the mouth rinse. The potential impact of this small difference in design should not be ignored. Lin and colleagues [32] found that a short rest break allows subjects to reallocate their neurocognitive resources during the subsequent work period. Consequently, the increased cognitive demand or increase in mental fatigue could occur without a decrement in cognitive performance. While the changes observed in cognitive performance during the SCWT in this investigation are consistent with other investigations, these results do not allow us to conclude that the mentally fatiguing task negatively impacted cognitive function.

In contrast, the changes in perceived mental fatigue observed in this investigation do suggest that we were able to facilitate mental fatigue. Specifically, we observed increased perceived mental fatigue in both conditions. However, the increase in mental fatigue during PLAC was greater than that observed in CHO. These findings are very similar to those of Van Cutsem and colleagues [33] who used a caffeine-maltodextrin mouth rinse. It is curious that the increase in mental fatigue was not associated with a decrease in cognitive function in this investigation. It is possible that the increase in mental fatigue was a consequence of a greater mental demand facilitated by prolonged engagement in the SCWT. The use of more cognitive resources during prolonged SCWT application could explain how performance during the SCWT could be maintained while the perception of mental fatigue was increased. In order to assess this possible outcome, it would have been appropriate to include a novel test that assessed cognitive function before and after the SCWT. The specific mechanism underlying the increased perception of mental fatigue following prolonged engagement in the SCWT is unclear. However, it could be related to cortical excitability. Therefore, the procedures used in this investigation were able to induce an increased perception of mental fatigue but did not alter measures associated with cognitive function.

In this investigation, we observed a 17% reduction in corticomotor excitability in PLAC after mental fatigue. Several investigators have documented a change in the EEG response to visual [34,35] and auditory [36] stimuli after mental fatigue. The findings in these investigations suggest that the reduction in cortical excitability is manifested by reduced amplitudes and prolonged latencies of event-related potential (ERP) components. Functionally these changes in EEG responses subsequent to mental fatigue are associated with reduced attention [35], the ability to ignore irrelevant stimuli [34], and motivation to complete the task [36].

There are only two other investigation we are aware of that evaluated the impact of mental fatigue on corticomotor excitability by using TMS [37,38]. While Rozand and colleagues reported that mental fatigue did not change corticomotor excitability, close examination of the research design used and the findings reported does not provide certainty that these authors were able to directly consider this question [38]. Specifically, corticomotor excitability was measured at rest and during pointing before and after a mentally fatiguing task. Unfortunately, the order of measurement was not consistent before, and after, the mental fatigue task, so the act of pointing could have altered resting or non-active measurement of corticomotor excitability. Morris and Christie also did not document a significant reduction in corticomotor excitability following mental fatigue, but the magnitude of reduction in corticomotor excitability (MEP amplitude) they observed (17%) is similar to that observed in our investigation [37]. These authors reported a change in cortical function during TMS, as described by a significant increase in cortical silent period (CSP), following mental fatigue.

The negative impact of mental fatigue on corticomotor excitability may be mediated through the anterior cingulate cortex (ACC). The SCWT and other mentally fatiguing tasks appear to activate the ACC [39] and mental fatigue has been associated with reduced dopaminergic transmission to the striatum and ACC [20]. While the findings reported here suggesting that corticomotor excitability is reduced following mental fatigue are consistent with investigations describing reductions in cortical excitability, it is evident that efforts need to be made to confirm our finding that mental fatigue negatively impacts corticomotor excitability.

This investigation was also intended to determine if a CHO mouth rinse could attenuate any negative impact of mental fatigue on corticomotor excitability. The negative impact of the mentally fatiguing task on corticomotor excitability was reduced by 13% when a CHO mouth rinse was applied. These findings are consistent with two previous investigations that examined the impact of a mouth rinse on corticomotor excitability using TMS stimulation of the FDI. Gant et al. found a 9% increase in MEP of the FDI in normal participants after using a carbohydrate mouth rinse [23]. In comparison, Gam et al. observed a 16% increase in MEP of the FDI in competitive cyclists after using a quinine mouth rinse [40]. Similarly, in a previous investigation, our laboratory found a 9% increase in the MEP of the quadriceps muscles after application of a CHO mouth rinse identical to the one used in this investigation [25]. Conceptually, a CHO mouth rinse could increase corticomotor excitability by stimulating receptors in the mouth that send afferent information to the sensorimotor cortex, effectively priming it for motor activity. The receptors in the mouth responsible for this activation may detect the presence of energy dense nutrients communicating to the motor cortex that energy is available for movement [23].

The interpretation of the results provided here becomes more complex when the magnitude of change in perceived mental fatigue is compared with MEP. In particular, perceived mental fatigue increased in both conditions, while MEP only decreased in PLAC. Consequently, concluding that changes in corticospinal activity are directly and proportionality related to changes in perceived mental fatigue is tenuous. The current investigation is not effectively constructed to directly address this question, suggesting that a properly powered future investigation examining the impact of a mentally fatiguing alone on corticospinal activity is warranted.

The methodology used in this investigation does not allow us to identify the specific locations or pathways that could mediate the observed phenomenon. However, Turner and colleagues [41] have previously described the areas of the brain that respond to a CHO mouth rinse during motor activity. Using fMRI, these investigators demonstrated that CHO in the mouth primed the regions in the cerebral cortex that subsequently increased activation within the sensorimotor cortex during a motor task. While the ACC was identified as one of the brain areas activated by CHO, numerous other areas were activated including but not limited to the precentral gyrus, the postcentral gyrus, the opercular cortex, the lateral occipital cortex, and the insular cortex [41]. Speculation of the brain areas involved in this phenomenon becomes more problematic when the work of Svane and colleagues is considered [42]. These authors suggest that most of the muscle activity initiated by TMS stimulation directed at the FDI is mediated via direct monosynaptic connections from the motor cortex to spinal motor neurons and there is little evidence that indirect pathways play significant role [42]. As a consequence, a mechanistic understanding of the relationships between mental fatigue, CHO stimulation of receptors in the oral cavity, and corticomotor excitability remain elusive and should be further examined.

There are two other important limitations to this experimental design that should be considered. This experimental design did not allow us to separate the impact of the mental fatigue precipitated by the SCWT from the effect of repetitive finger movement during the task. Exhaustive unilateral muscle contractions have been shown to alter MEP in contralateral homologous muscle [43,44,45]. While finger movement during the SCWT is not considered exhaustive or fatiguing, it is possible that the changes observed here were due to finger movement, as well as mental fatigue. Future investigations in this area should be designed to differentiate between mental fatigue and repetitive movement. Another limitation of this investigation is lack of inclusion of a physical performance measure. The complexity of measurement of corticomotor excitability makes the concomitant measurement of physical performance difficult. However, a change in corticomotor excitability without a change in physical performance may be pragmatically inconsequential and requires further investigation.

## 5. Conclusions

In conclusion, the results of this investigation demonstrate that an increased perception of mental fatigue is associated with a reduction in corticomotor excitability. Furthermore, it appears that a CHO mouth rinse attenuates the negative impacts of a mentally fatiguing task on perceived fatigue and corticomotor excitability. Interestingly, the CHO mouth rinse did not influence cognitive function during the task. The results of this investigation suggest that it may be valuable to examine the impact of a CHO mouth rinse on corticomotor function subsequent to mental fatigue and the potential impact of physical performance.

## Figures and Tables

**Figure 1 brainsci-11-00972-f001:**
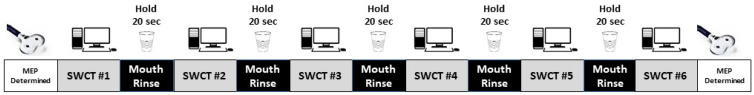
Experimental protocol to determine motor evoked potential (MEP) before and after repeated Stroop Color Word Test (SCWT).

**Figure 2 brainsci-11-00972-f002:**
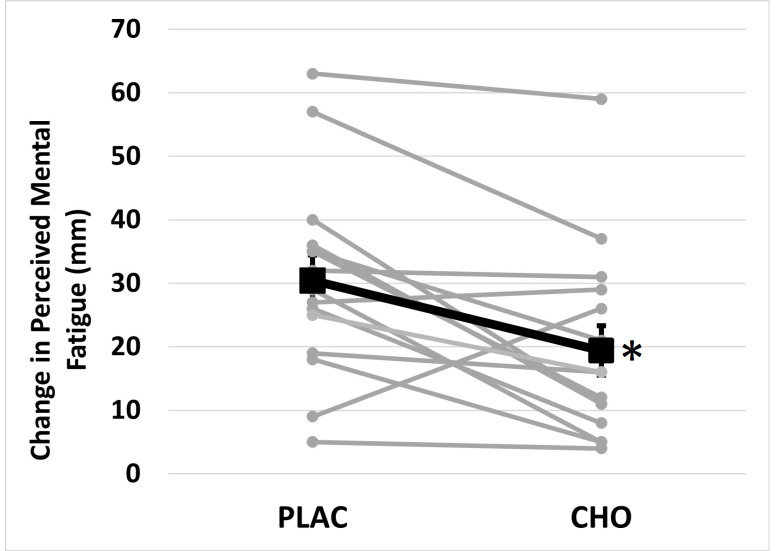
Change in perceived mental fatigue following Stroop Color Word Test (SCWT) when using placebo (PLAC) or (CHO) mouth rinse. * Indicates difference (*p* = 0.005) from PLAC.

**Figure 3 brainsci-11-00972-f003:**
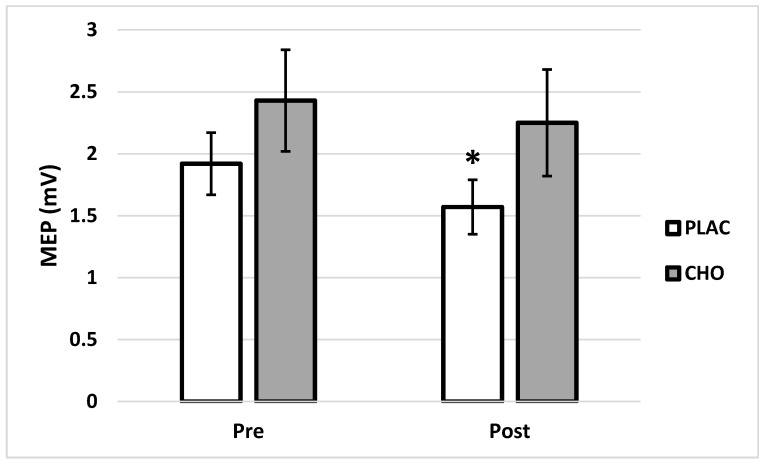
Absolute motor evoked potential (MEP) before and after Stroop Color Word Test (SCWT) when using (PLAC) or carbohydrate (CHO) mouth rinse. * Indicates difference (*p* = 0.019) from Pre.

**Figure 4 brainsci-11-00972-f004:**
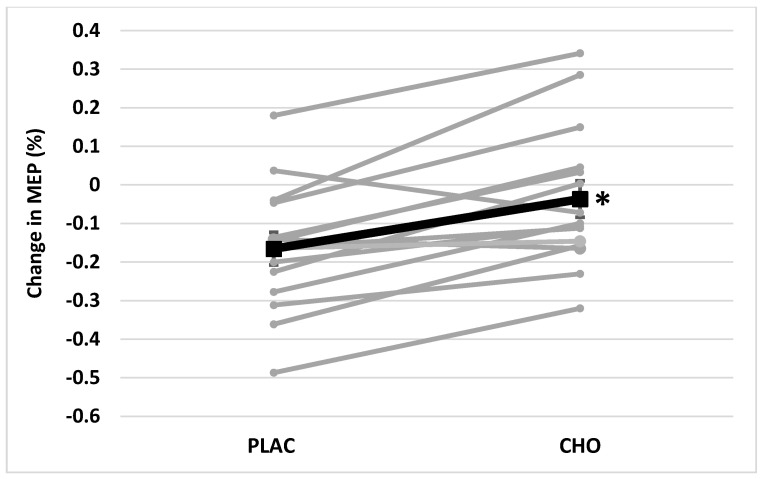
Relative change in motor evoked potential (MEP) after Stroop Color Word Test (SCWT) when using (PLAC) or carbohydrate (CHO) mouth rinse. * Indicated difference (*p* < 0.001) from PLAC.

**Table 1 brainsci-11-00972-t001:** SWCT performance over repeated application when using placebo (PLAC) or carbohydrate (CHO) mouth rinse.

Block		Congruent Reaction Time (ms) *	Incongruent Reaction Time (ms) *	Control Reaction Time (ms) *	Congruent Accuracy (%)	Incongruent Accuracy (%)	Control Accuracy (%)
1	PLAC	777 ± 26	830 ± 29	783 ± 28	97.3 ± 0.7	96.3 ± 0.7	97.5 ± 0.6
	CHO	749 ± 27	809 ± 37	757 ± 28	97.6 ± 0.6	97.6 ± 0.5	97.4 ± 0.9
2	PLAC	745 ± 21	789 ± 23	749 ± 20	96.8 ± 0.9	97.2 ± 0.8	97.3 ± 0.8
	CHO	711 ± 21	766 ± 28	729 ± 28	97.8 ± 0.8	96.2 ± 0.8	97.3 ± 0.6
3	PLAC	742 ± 21	783 ± 21	739 ± 20	97.5 ± 0.7	95.9 ± 1.0	97.0 ± 0.7
	CHO	703 ± 14	746 ± 22	698 ± 15	97.3 ± 0.9	97.3 ± 0.7	96.9 ± 0.8
4	PLAC	741 ± 21	791 ± 20	734 ± 18	96.6 ± 0.8	95.6 ± 1.0	97.2 ± 0.8
	CHO	690 ± 15	743 ± 20	694 ± 19	96.8 ± 0.6	95.6 ± 0.9	97.1 ± 0.6
5	PLAC	731 ± 21	772 ± 23	730 ± 19	97.0 ± 0.8	95.7 ± 0.9	97.4 ± 0.8
	CHO	690 ± 14	741 ± 25	688 ± 18	97.4 ± 0.6	96.1 ± 0.9	97.4 ± 0.5
6	PLAC	720 ± 22	770 ± 23	724 ± 21	97.3 ± 0.7	95.4 ± 0.7	97.6 ± 0.8
	CHO	678 ± 17	721 ± 20	690 ± 17	97.2 ± 0.7	96.3 ± 0.8	97.1 ± 0.7

* Indicates difference across time (*p* < 0.01).

**Table 2 brainsci-11-00972-t002:** Perceived mental fatigue before and after repeated SWCT when using (PLAC) or carbohydrate (CHO) mouth rinse.

	Pre (mm)	Post (mm)
PLAC	14 ± 3	45 ± 3 *
CHO	18 ± 4	37 ± 5 *

* Indicates difference from Pre.

## Data Availability

This data is available on FigShare at https://doi.org/10.6084/m9.figshare.14766354.

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
