# Peer review of "Impact of a Carbohydrate Mouth Rinse on Corticomotor Excitability after Mental Fatigue in Healthy College-Aged Subjects"

_brainsci, 2021, doi:10.3390/brainsci11080972_

Round 1
Reviewer 1 Report
A brief summary
The present study suggests that perceived mental fatigue elicited by intermittent Stroop Color Word Test (SCWT), estimated by visual analog scale, may depress corticomotor excitability, measured by single transcranial magnetic stimulation (TMS) to the motor hand area, and that periodical carbohydrate mouse rinse (CHO MR) during the SCWT may attenuate the perceived mental fatigue and depressed corticomotor excitability after the SCWT.
Broad comments
While the study seemed to address a valuable intervention to attenuate mental fatigue, the article lacks scientific rationale to make a hypothesis and the data and analyses are not presented appropriately. In my opinion, the research design is inappropriate to elucidate relationship between mental fatigue and corticomotor excitability of hand.
Specific comments
Lines 46-55
To my knowledge, only one study (Pageaux et al. 2015 Front Hum Neurosci) have reported an increased EMG activity to maintain power output during constant-intensity whole body exercise after mental fatigue. The hypothesis of this study based on premise that mental fatigue results in an increased motor command in order to maintain output is not rational.
Lines 56-61
As authors stated, corticomotor excitability is altered by motor task or skill training. However, the argument that a reduced physical performance is related to a depressed corticomotor excitability in resting muscle measured by single TMS is simplistic. It have been demonstrated that a decreased MEPs due to fatiguing muscle contractions was not associated with an impairment of MVC after muscle fatigue (Gandevia et al. 1996 J Physiol).
Lines 72-79
Since CHO MR has been shown to enhance corticomotor excitability, it is not clear the major difference of this present study with the previous article. A rationale of applying CHO MR to mental fatigue as intervention should be described.
Lines 172-173
How many trials were discarded?
Lines 175-186
Why were control condition, in which simple finger tapping was repeated for 60 min, performed? Repeated contraction of left fingers during SCWT can alter corticomotor excitability of right hand area despite of effect of SCWT on MEPs (Aboodarda et al. 2016 Scand J Med Sci Sport; Bonato et al. 1996 Neurosci Lett; Takahashi et al. 2009 Clin Neurophysiol).
Line 183
What is control stimuli?
Lines 223-225
For mean absolute perceived mental fatigue, authors should show simple main effect (PLAC vs. CHO) at Pre and Post.
Lines 235-237
Why was log transformed MEP data shown in figure?
Line 239
For mean MEP, authors should show simple main effect (PLAC vs. CHO) at Pre and Post. In Figure 3, was there significant difference between PLAC and MDX at Pre? This point is important for interpretation of MEP results since effects of intervention on MEP is dependent on MEP size before the intervention.
Lines 324-328
In the CHO MR (MDX) condition, perceived mental fatigue was significantly increased after the SCWT even though CHO MR was applied. If perceived mental fatigue is relate to a decreased corticomotor excitability, MEP should have been decreased in the CHO condition. Authors should explain this discrepancy. Furthermore, the article did not discuss whether the degree of a depressed corticomotor excitability is dependent on the degree of perceived mental fatigue.
Lines 363-369
In Introduction, authors stated relationship between a reduced physical performance due to mental fatigue and changes in corticomotor excitability. However, it is unclear whether an enhanced corticomotor excitability by CHO MR influences physical performance. Therefore, functional significance for applying CHO MR is not very clear to me.
Reviewer 2 Report
The methods section is very complete and easy to understand. TMS protocols are appropriate. Can the authors state why they excluded subjects who had received non-invasive brain stimulation in the past year - what is the justification for this. If remote stimulation has an effect, would not the experimental protocol have a greater potential effect? do the authors have any data to document the reliability of their MEP amplitudes with their method?
Results:
The legend in Figures 2-4 reflecting the mouth rinse does not match with the graph. Legend states CHO while figure shows MDX. Please change so these are congruent for all figures.
Discussion: The authors adequately explain the impact of mental fatigue on MEP amplitude. Where the argument becomes less convincing is the incongruency between the mental fatigue and cognitive performance. They are correct instating that their protocol did not induce a reduction in cognitive performance. Can the authors comment on potential learning effects during the task that may be masking cognitive performance challenges to fatigue. It would be valuable to consider data after which the rate of reaction time reduction has plateaued or reversed to better evaluate cognitive performance on corticomotor excitability.
Round 2
Reviewer 1 Report
The authors have improved the manuscript. However, there are still two major concerns about rationale for hypothesis and experimental design.
I agree that mental fatigue results in greater perceived exertion during constant intensity exercise. The greater perceived exertion may be related to increased motor command during exercise. But whether the greater exertion and the increased motor command are attributed to a decreased corticomotor excitability is unclear. When corticomotor excitability is decreased, is greater motor command required to maintain constant force? If so, the authors should add references.
The authors argue that a decrease in MEP after SCWT was associated to mental fatigue. However, I disagree this argument since effects of right finger tapping alone on MEP in left FDI was not determined in this study. It have reported that unilateral muscle contractions not only increased MEP in the contralateral homologous muscle but also decreased that (Bonato et al. 1996; Takahashi et al. 2009). Therefore, the results of the this manuscript can not exclude a possibility that the decreased MEP after SCWT was elicited by right finger tapping alone.
Author Response
Thank you for these comments. We have made another concerted effort to address your concerns and hope they are acceptable.
The authors have improved the manuscript. However, there are still two major concerns about rationale for hypothesis and experimental design.
I agree that mental fatigue results in greater perceived exertion during constant intensity exercise. The greater perceived exertion may be related to increased motor command during exercise. But whether the greater exertion and the increased motor command are attributed to a decreased corticomotor excitability is unclear. When corticomotor excitability is decreased, is greater motor command required to maintain constant force? If so, the authors should add references.
To date we are not aware of any publications that have manipulated corticomtor excitability and observed the change in motor command. We have included references in the introduction demonstrating the relationships that have been observed in changes in brain activity of the motor cortex and corticospinal activity. We feel it is important to remember that “motor command” is a conceptual idea that is used differently in the literature. In an effort to minimize any confusion about our investigation we have avoided making an conclusion or having and discussion about “motor command” and limited our conclusion and discussion to the measures used in this investigation.
The authors argue that a decrease in MEP after SCWT was associated to mental fatigue. However, I disagree this argument since effects of right finger tapping alone on MEP in left FDI was not determined in this study. It have reported that unilateral muscle contractions not only increased MEP in the contralateral homologous muscle but also decreased that (Bonato et al. 1996; Takahashi et al. 2009). Therefore, the results of the this manuscript can not exclude a possibility that the decreased MEP after SCWT was elicited by right finger tapping alone.
We recognize the possibility that finger movement could include influence MEP in contralateral homologous muscle and have included this as a limitation of the study. While we feel that equivocating the fatigue activities to the activity of finger tapping during the SCWT is tenuous, we recognize this possibility can not be ruled out. We hope our additive language addresses this concern for the reviewer.
Reviewer 2 Report
Thank you for your reply to the points raised in the initial review. Two further edits are suggested: 1) the CHO/MDX labels were changed on some portions of the figures but not others, making interpretation confusing. Is MDX supposed to be included in the figures? If so, I don't see any reference to that abbreviation in the text. 2) the authors' statement on why individuals with a history of non-invasive stimulation in the past year were excluded was very helpful (as were the MEP reliability data). I would suggest a brief statement to that effect in the text for clarity.
Author Response
Thank you for these comments. We hope the responses below address your concerns.
Reviewer 2
Thank you for your reply to the points raised in the initial review. Two further edits are suggested: 1) the CHO/MDX labels were changed on some portions of the figures but not others, making interpretation confusing. Is MDX supposed to be included in the figures? If so, I don't see any reference to that abbreviation in the text. 2) the authors' statement on why individuals with a history of non-invasive stimulation in the past year were excluded was very helpful (as were the MEP reliability data). I would suggest a brief statement to that effect in the text for clarity.
We have examined the document closely and can not find any use of MDX in the current document including the labels for figures. If we have missed any of these labels it would be helpful for the reviewer to provide a specific location.
We have edited the text from the previous version to “…treated clinically with non-invasive brain stimulation for a brain or psychological disorder within the past year..”. We hope this meets the reviewer’s concern. After review of our included and excluded subjects, we did not exclude anyone for this particular purpose. The specific criteria was included in our IRB (ethics) application and was used based on the recommendation of another researcher who uses TMS, so we think it is appropriate to include it in the description.